# Long-Term Outcome of Neoadjuvant Tyrosine Kinase Inhibitors Followed by Complete Surgery in Locally Advanced Dermatofibrosarcoma Protuberans

**DOI:** 10.3390/cancers13092224

**Published:** 2021-05-06

**Authors:** Jessica Beaziz, Maxime Battistella, Julie Delyon, Cécile Farges, Oren Marco, Cécile Pages, Christine Le Maignan, Laetitia Da Meda, Nicole Basset-Seguin, Matthieu Resche-Rigon, Anouk Walter Petrich, Delphine Kérob, Céleste Lebbé, Barouyr Baroudjian

**Affiliations:** 1Department of Dermatology, AP-HP Saint-Louis Hospital, F-75010 Paris, France; julie.delyon@aphp.fr (J.D.); laetitia.da-meda@aphp.fr (L.D.M.); nicole.basset-seguin@aphp.fr (N.B.-S.); delphine.kerob@aphp.fr (D.K.); celeste.lebbe@aphp.fr (C.L.); barouyr.baroudjian@aphp.fr (B.B.); 2Department of Pathology, AP-HP Saint-Louis Hospital, F-75010 Paris, France; maxime.battistella@aphp.fr; 3INSERM Unity 976, Team 1, Université de Paris, HIPI, F-75010 Paris, France; 4Department of Radiology, AP-HP Saint-Louis Hospital, F-75010 Paris, France; cecile.farges@gmail.com; 5Department of Reconstructive Surgery, Saint-Louis Hospital, F-75010 Paris, France; oren.marco@aphp.fr; 6Department of Dermatology, Paul-Sabatier Toulouse III University, CHU de Toulouse, 31059 Toulouse, France; pageslaurent.cecile@iuct-oncopole.fr; 7Department of Oncology, AP-HP Saint-Louis Hospital, F-75010 Paris, France; christine.lemaignan@aphp.fr; 8Biostatistics and Medical Information Department, AP-HP Saint-Louis Hospital, F-75010 Paris, France; matthieu.resche-rigon@u-paris.fr (M.R.-R.); anouk.walter-petrich@aphp.fr (A.W.P.)

**Keywords:** dermatofibrosarcoma protuberans, imatinib, tyrosine kinase inhibitor, long-term, neoadjuvant

## Abstract

**Simple Summary:**

Wide surgical excision is the standard treatment for dermatofibrosarcoma protuberans. Imatinib mesylate has been reported as an efficient neoadjuvant therapy to surgery in order to reduce tumor size and post-operative relapses for locally advanced or unresectable tumors. The aim of this study was to evaluate the long-term status of patients with advanced dermatofibrosarcoma protuberans treated by neoadjuvant tyrosine kinase inhibitors. Based on the data of 27 patients in our center, locally advanced and unresectable DFSP were efficiently treated with neoadjuvant tyrosine kinase inhibitors followed by complete surgery with micrographic analysis with durable local recurrence disease-free survival and few severe adverse events.

**Abstract:**

In locally advanced dermatofibrosarcoma protuberans (DFSP), imatinib mesylate has been described as an efficient neoadjuvant therapy. This retrospective study included patients with locally advanced DFSP who received neoadjuvant TKI (imatinib or pazopanib) from 2007 to 2017 at Saint Louis Hospital, Paris. The primary endpoint was the evaluation of the long-term status. A total of 27 patients were included, of whom nine had fibrosarcomatous transformation. The median duration of treatment was 7 months. The best response to TKI treatment before surgery, evaluated according to RECIST1.1 on MRI, consisted of complete/partial response (38.5%) or stability (46.2%). DFSP was surgically removed in 24 (89%) patients. A total of 23 patients (85%) were disease-free after 64.8 months of median follow-up (95% confidence interval 47.8; 109.3). One patient developed distant metastases 37 months after surgical tumor resection and finally died. Two patients (7%) did not get surgery because of metastatic progression during TKI treatment, and one patient refused surgery even though the tumor decreased by 30%. Treatment-related adverse events (AE) occurred in 23 patients (85%). Only four patients (imatinib: *n* = 3, pazopanib: *n* = 1) had grade ≥3 AE requiring temporary treatment disruption. Neoadjuvant TKI followed by complete surgery with micrographic analysis is an effective strategy for locally advanced and unresectable DFSP, with durable local recurrence disease-free survival.

## 1. Introduction

Dermatofibrosarcoma protuberans (DFSP) is a rare soft-tissue sarcoma [1] representing approximatively 6% of all soft-tissue sarcomas and characterized by slow infiltrative growth of CD34+ spindle cells with high risk of local recurrence and low rate of metastasis [2]. Although most tumors are low-grade, a high-grade fibrosarcomatous component is found in 10–20% of cases [3].

The standard treatment of localized primary or recurrent cases is wide excision to obtain tumor-free margins. A complete surgery with micrographic analysis of margins reduces the local recurrence rate to less than 10% [4,5]. However, in some cases, the first resection is not possible due to tumor extension or size, or the risks of cosmetic or functional impairment [6]. Metastatic disease and local relapse after a complete surgical resection are closely related to fibrosarcomatous transformation [7,8,9].

DFSP is characterized by a genomic rearrangement involving chromosomes 17 and 22, in a supernumerary ring chromosome containing chromosomes 17 and 22 sequences, or in a reciprocal balanced translocation t (17; 22) (q22; q13). It leads to the fusion of collagen 1 alpha 1 (COL1A1) and platelet-derived growth factor beta (PDGFβ) genes [3,10,11,12]. The result of this rearrangement is an abnormal activation through an autocrine loop of the PDGFβ, which is a potent mitogen for mesenchymal cells that activates the Ras MAPK and PI3K–AKT–mTOR pathways, leading to cell growth and differentiation [11,13,14]. The identification of dysregulated expression of PDGFβ led to the use of tyrosine kinase inhibitors (TKI), such as imatinib mesylate, to treat patients with unresectable, recurrent, or metastatic DFSP [15,16]. Several studies have shown the benefit of neoadjuvant imatinib to reduce tumor size, facilitating complete surgical resection and decreasing early post-operative relapses [12,17,18,19].

Although imatinib is the main TKI for treatment of DFSP, pazopanib could be used in case of failure of imatinib in unresectable tumors. It is a multityrosine kinase inhibitor (VEGF, PDGF, KIT) with anti-angiogenic activity, which is also approved in soft-tissue sarcomas [20,21,22,23]. Given the rarity of this tumor, there are very few reports of a cohort of DFSP patients treated with imatinib as a neoadjuvant treatment, with a long-term follow up [24].

The aim of this study was to evaluate the long-term outcome of patients with locally advanced DFSP treated by neoadjuvant TKI (imatinib or pazopanib).

## 2. Materials and Methods

### 2.1. Study Design

We conducted a retrospective monocentric study of all patients with histologically proven locally advanced DFSP at Saint-Louis Hospital Paris, who received neoadjuvant TKI (imatinib, pazopanib) from November 2007 to February 2017. Patients were selected from the multidisciplinary tumor board records. The inclusion criteria were locally advanced and unresectable, primary, or recurrent DFSP, without metastasis, and confirmed by pathological examination. The first surgical resection was not feasible because of aesthetic or functional risks, such as head and neck tumors, locoregional invasion of underlying structures: tendon, muscle, bone, or periosteum. Non-resectability criteria were also appreciated by the surgeon who evaluated that he could not perform complete surgery without sequelae. Hence, these patients were eligible for a neoadjuvant treatment. All the patients had cytogenetic confirmation of the presence of COL1A-PDGFβ (by fluorescence in situ hybridization (FISH)). This study was approved by the local ethical committee. Informed written consent was obtained for all patients.

### 2.2. Treatment

The neoadjuvant TKI treatment consisted of imatinib mesylate 400 to 800 mg once daily depending on the tolerance, or pazopanib 600 to 800 mg daily. In case of grade 3 or 4 toxicities, the treatment was interrupted and resumed at a lower dose after adverse event resolution (≤grade 1). The switch from one to the other TKI was authorized in case of limited toxicity or lack of efficacy (progression disease according to RECIST 1.1) as well as the switch to another TKI (sunitinib, nilotinib, two multityrosine kinase inhibitors targeting the PDGR receptor) in case of progression disease according to RECIST 1.1. The first tumor evaluation after TKI introduction was performed at month 3 and then every 3 months until surgery was performed. No maximum duration of treatment was planned. Feasibility of resection, depending of radiological response, was discussed regularly in a multidisciplinary board comprising an experimented surgeon and radiologist. Adjuvant radiotherapy was optional and was discussed for each patient individually, depending on tumor size and fibrosarcomatous transformation status.

### 2.3. Endpoints

The primary endpoint of this study was the long-term status from the date of complete tumor resection to the date of relapse (local or metastasis), death due to any cause, or the most recent follow-up. The follow-up after surgery consisted of clinical examination with local MRI every six months and associated with a chest CT scan for patients with transformation. Secondary endpoints included radiological response before the surgery using RECIST 1.1 criteria, histological response based on surgical specimen analysis, and safety using the Common Terminology Criteria for Adverse Events v5.0.

### 2.4. Pathological and Immuno-Histochemical Examination

Pre-TKI biopsies and post-TKI surgical resection specimens were retrospectively analyzed by a trained dermatopathologist. Analysis performed on the biopsy samples evaluated mitotic index, CD34 staining, and fibrosarcomatous transformation. Maximal histological diameter and thickness were studied on the surgical specimens. Therapeutic response was evaluated by the percentage of the surface response, the location of this response with the deepest anatomic structure invaded, the density of tumor-infiltrating lymphocytes (TILs), and mitotic index post TKI.

## 3. Results

### 3.1. Patients

Patients characteristics at baseline are summarized in Table 1. Twenty-eight patients received TKI as neo-adjuvant therapy during the study period. One patient was excluded from the analysis because of lung metastatic disease that was retrospectively present before inclusion. A total of 27 patients were included in the study, of whom nine had pathological fibrosarcomatous transformation on pre-TKI biopsy. The COL1A-PDGFB fusion gene was detected in all patients using fluorescence in situ hybridization technique. There was a predominance of men (56%). Median tumor size (longest diameter determined by clinical assessment) was 6.0 cm (interquartile range (IQR) = (5.0; 11.3)). The median age at the beginning of the TKI treatment was 45.1 years old (IQR (38.1; 49.6)) (Table 1). Lesions were localized on the trunk (*n* = 16), scalp/face (*n* = 9), and limbs (*n* = 2). Locoregional invasion of underlying structures were present in 19 patients: tendon (*n* = 3)/muscular (*n* = 9), bone abutment (*n* = 5)/periosteum (*n* = 2). Seventeen patients had been previously treated by large but incomplete surgical resection before receiving TKI treatment because the diagnosis of DFSP was not initially suspected and clinically relapsed.

### 3.2. Treatment

Twenty-two patients received imatinib (median dose = 600 mg/day), two received pazopanib (median dose = 700 mg/day), and three received several TKI due to side effects or inefficiency of the first TKI before surgery (Table 1). For one patient, imatinib was replaced with pazopanib because of cholestasis, and for two patients, imatinib was replaced respectively with pazopanib or sunitinib and nilotinib because of progression with imatinib. Median treatment duration was 7 months (IQR (4.61; 12.48)).

DFSP was surgically removed in 24 patients (89%) after a median follow-up of 7 months. Two patients (7%) did not get surgery because of metastatic progression during TKI treatment (respectively 14 and 11 months after starting TKI), and one patient refused surgery even though the tumor decreased by 30% and was lost during follow-up.

The first surgery was performed within 2 weeks after TKI withdrawal. The initial surgical margins were delineated based on the border of the palpable lesions and were comprised between 10 and 20 mm, which were chosen according to the localization, the skin laxity, and the size of the tumor margins. Pathological analysis used the ‘vertical modified technique’. The first resection was complete for 10 patients (42%), but additional surgical resections were needed to obtain clear margins for other patients (10 patients with two resections and four patients with three or more resections). Three patients received adjuvant radiation therapy completing surgical resection, as two of them presented tumor progression under neoadjuvant imatinib requiring its switch with another TKI. The third patient received adjuvant radiotherapy because the primary tumor was larger than 20 cm with a fibrosarcomatous transformation (Table 1).

### 3.3. Efficacy

Twenty-three patients (85%) were disease-free after a median follow-up of 64.8 months (IC at 95%: (47.8; 109.3)). One patient developed several metastases (subcutaneous, pulmonary, pleural, lymph nodes, adrenal glands) 37 months after the complete surgical resection, with no local relapse, and they died after 8 years. Concerning the three patients who did not get surgery, two patients presented metastatic progression during TKI but were alive at the end of the study, and the third patient presented a stability disease under TKI and was lost during follow-up. The best response to TKI treatment before the surgery evaluated on MRI consisted of complete response (CR) in 7% (two of 26), partial response (PR) in 30% (eight of 26), stable disease (SD) in 46% (12 of 26), or progression disease (PD) in 15% (four of 26). Thus, two patients with radiologically PD had finally complete surgery requiring wide resection and skin grafting. No imaging data were available in the records for one patient. The disease control rate (CR, PR, and SD) was 85%. Pre and post-MRI evaluation showed a median decrease of 24% (IQR (−0.38; 0)) of the maximal tumor diameter when compared to initial tumor diameter (Figure 1, Table 2). Among the patients with fibrosarcomatous DFSP, one patient (11%) had CR, one patient (11%) had PR, four patients (44%) had SD, and two patients (22%) had PD.

### 3.4. Histological Analysis

Among the 24 patients who had surgery, 23 had histological analysis in our center. Histopathological response was characterized by a decrease in a cellular density and modifications of extracellular components with varying degrees of hyalinic fibrosis. Localization of the histopathological therapeutic response was patchy (*n* = 10), diffuse (*n* = 7), central (*n* = 4), peripheral (*n* = 1), or absent (*n* = 1). The median percentage of therapeutic response surface was 65% (range 0–95%). The tumor-infiltrating lymphocytes after TKI were absent in 1 (4%), non-brisk in 21 (91%), and brisk in 1 (4%) patients (Figure 2). CD34 was expressed in all surgical specimens after TKI treatment. Median maximum tumor thickness was 16 mm (range 5–55). The last anatomic structure invaded by the tumor was subcutaneous fat in eight cases, sub-aponeurotic adipose tissue in seven cases, and striated skeletal muscle in eight cases. Before TKI treatment, the median mitotic index on initial biopsy (including the patients with progression before surgery) was 4. After TKI treatment, the median mitotic index, evaluated on surgical specimen was reduced to 0. On average, the mitotic index decreased after TKI by 60% (range: −100%–+60%) (Figure 1 and Figure 2).

### 3.5. Safety

Treatment-related adverse events occurred in 23 patients (85%). These were asthenia (grade 1 or 2, *n* = 14), edema (grade 1, *n* = 9), diarrhea (grade 1 or 2, *n* = 4), abdominal pain (grade 1, *n* = 3), muscle pain (grade 1, *n* = 2), acid reflux (grade 1, *n* = 1), aphthous stomatitis (grade 1, *n* = 1), thrombopenia (grade 1, *n* = 1), tachycardia (grade 1, *n* = 1), transient memory loss (grade 1, *n* = 1), CPK increased (grade 2, *n* = 1), and transaminitis (grade 1 *n* = 1). Four patients (17%) (imatinib: *n* = 3, pazopanib: *n* = 1) had grade 3 or higher toxicities. Two patients had grade 3 or 4 neutropenia, requiring temporary treatment discontinuation for one month and subsequent dose reduction. One of these two patients also presented grade 3 anorexia with significant weight loss (ten kilos in 1 year). Another patient presented grade 3 nausea requiring treatment disruption for one week and dose reduction. One patient presented grade 3 cholestasis with pazopanib 400 mg/d requiring treatment discontinuation and a switch to imatinib. Cholestasis was controlled after treatment interruption with no recurrence after imatinib replacement (Table 3).

## 4. Discussion

To our best knowledge, this study presents the largest cohort of locally advanced DFSP without metastases and treated with neoadjuvant TKI with a long-term follow-up. Only one patient presented with distant recurrence after complete surgery without local recurrence. Metastatic DFSP was rare.

As shown by Kérob et al. [17], this study confirms that TKI and especially imatinib mesylate are effective neoadjuvant treatments for DFSP. The first purpose of our study was to confirm these results at long-term follow-up and in particular the efficacity of both imatinib and pazopanib on long-term disease-free survival. We included all patients who received tyrosine kinase inhibitors with a neoadjuvant intention, whether or not they got surgery and not only present data of patients with a good response to TKI allowing surgery. In addition to patients’ long-term status, we analyzed clinical response, radiological response before surgery, and histological response based on surgical specimen. After 50 months of follow-up, 85% of our patients were disease-free.

The long-term efficacy of imatinib was previously published including inoperable but also metastatic patients. In 2005, McArthur et al. described eight patients with locally advanced or metastatic DFSP treated with imatinib, with a duration of follow-up ranging from 32 to 845 days [12]. Rutkowski et al. reported in 2017 the long-term efficacy of imatinib in 31 patients (inoperable or metastatic) with a 5-year PFS rate of 58% [24]. These results are comparable to previous studies reporting the efficacy of imatinib as neoadjuvant therapy [25,26,27,28]. Several phase II trials also reported the benefit of imatinib in patients with advanced or metastatic DFSP, in order to reduce the extent of surgery [5,12,17,18,29]. To our knowledge, we present the first study assessing the efficacy and the safety of TKI, using both imatinib and pazopanib, as a neoadjuvant treatment for locally advanced and nonresectable DFSP with 5 years follow-up, in a large cohort.

In the present study, four patients received pazopanib, including one patient who had previously been treated with imatinib. The multicenter phase II study recently published reported only patients treated with pazopanib with 22% of tumor responses for patients with unresectable DFSP [22]. Pazopanib seems to induce a lower response compared to those previously reported for imatinib. In our study, three patients with pazopanib as first line were stable before surgery, but the low number of patients with pazopanib makes the comparison with imatinib impossible. One patient presented grade 3 cholestasis with pazopanib 400 mg/d requiring treatment discontinuation and a switch to imatinib. Delyon et al. also reported a low tolerance of pazopanib with 100% of grade 2 clinical or biological adverse events, and 74% of grades 3 and 4 adverse events. Our study concurs with the supposition that pazopanib should be considered in second-line therapy for DFSP, especially in resistance to imatinib, because of its response rate and toxicity profile.

Nine patients in our cohort had fibrosarcomatous transformation, and six of them presented a complete response or stability during the follow-up after the surgery. Sarcomatous transformation and large initial tumor size are the main risk factors for recurrent lesions and metastasis DFSP reported in the literature [7,30,31]. All our patients with metastatic progression before or after surgery had a fibrosarcomatous transformation. For one patient with transformation, the tumor locally relapsed 4 years before needing imatinib replacement with sunitinib 37.5 mg/d and then nilotinib 400 mg twice a day with partial efficacy. Finally, 3 years after complete surgery, distant metastasis developed, leading to the patient’s death. Another two patients with transformation presented metastatic progression, leading to imatinib discontinuation and replacement with pazopanib. None of them responded to this second-line TKI, and finally, they did not get surgery because of metastatic progression. Furthermore, for these three patients, the initial tumor size was large > 20 cm. Fibrosarcomatous transformation is also associated with partial response to imatinib. For tumors with fibrosarcomatous component or with poor prognostic markers, radiation therapy has been reported to improve local control and reduces the risk of recurrence after surgery [32,33].

Little is known about the mechanism of resistance to imatinib [34]. The action of imatinib is related to the inhibition of platelet-derived growth factor receptor beta (PDGFRB) [35]. It blocks cellular proliferation and induces the apoptosis of malignant cells and hyalinic fibrosis [36]. All of our patients presented COL1A–PDGFB fusion. The absence of response to imatinib could be associated to a weak PDGFRB phosphorylation [5] or a CDKN2A/p16 loss, thus implying CDK4 as a possible target in imatinib-resistant DFSP [37].

Another strength of this study is the pathological evaluation after TKI treatment. We sought to evaluate the effect of TKI on histologic parameters. Histological response was defined as decreased tumor density, mitotic index, and hyalinic fibrosis induction. The histological response after treatment can be diffuse, patchy, observed at the periphery or in the center of the tumor. In our study, most therapeutic responses were patchy or diffuse. There was no correlation between the localization of therapeutic response and clinical response. The only factor of poor prognosis was the absence of histologic response. Clinical response was previously correlated with decreased cell density, which was itself associated with the score for fibrosis [17]. Stacchiotti et al. also reported these pathological changes associated with response to imatinib in metastatic DFSP [34]. TILs play an important role in regulating tumor immunity. It has been demonstrated that imatinib modifies the immune profile of DFSP tumors, with activation of the T cells response at the tumor site. T cells infiltration was correlated with the pathological response. The presence of active T cells infiltration was required to progress from IM-induced tumor senescence to tumor apoptotic death [38]. The presence of TILs is a prognostic factor, and a dense lymphocytic infiltrate within and around the tumor is correlated with better prognosis in DFSP. Infiltrate with TILs was classified as absent, non-brisk, or brisk. These immunological changes in the tumor suggest that combined or sequential treatments with imatinib and immunotherapy could be a therapeutic option for advanced DFSP. Interestingly, in this study, only one patient who had a complete resection of DFSP with fibrosarcomatous transformation relapsed. This patient was the only one with no TILs and no pathological response. Despite adjuvant radiotherapy, the tumor relapsed after four years. This is consistent with the fact that the absence of lymphocytes infiltration in the tumor represents a risk factor of tumor progression in DFSP.

The adverse events reported in our study are consistent with what was previously shown, and they are dose dependent [17,18,39]. These side effects were mild for a vast majority of patients (83%). Grade 3 and 4 toxicities occurred in only four patients. Two of them received imatinib 800 mg per day, and the symptoms were easily managed by dose reduction or interruption.

This study has limitations, particularly due to the retrospective design. However, our data are consistent with previous studies concerning TKI efficacy in a neoadjuvant setting.

In order to reduce bias, we included all the patients who received TKI with a neoadjuvant intention, even if they finally did not get surgery.

## 5. Conclusions

Our retrospective study suggests the long-term efficacy of imatinib and pazopanib as a neoadjuvant therapy followed by completed surgery for locally advanced and unresectable DFSP, including for DFSP with transformation.

## Figures and Tables

**Figure 1 cancers-13-02224-f001:**
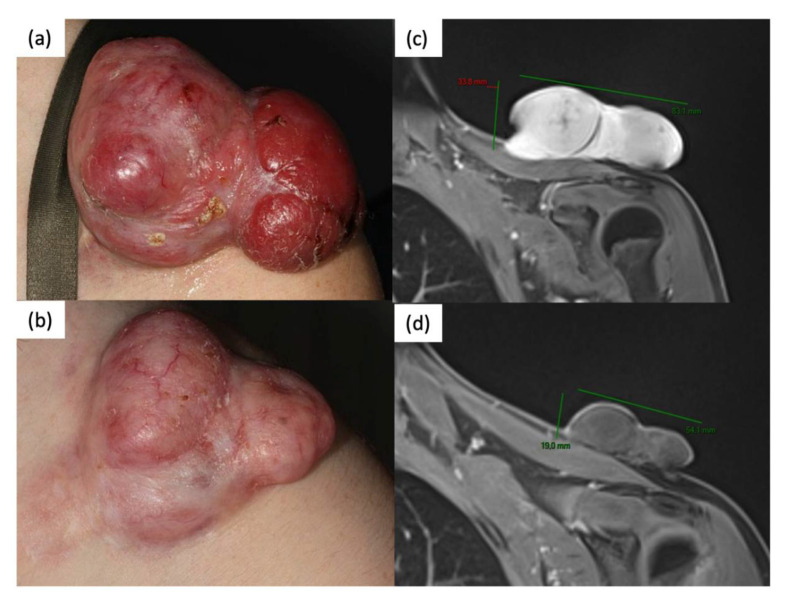
Patient with large DFSP located on the shoulder at baseline (**a**). Magnetic resonance imaging prior to the beginning treatment with imatinib mesylate (**c**). After 4 months therapy with imatinib, we observed a reduction of the tumor size (**b**) and of the signal intensity on MRI (**d**).

**Figure 2 cancers-13-02224-f002:**
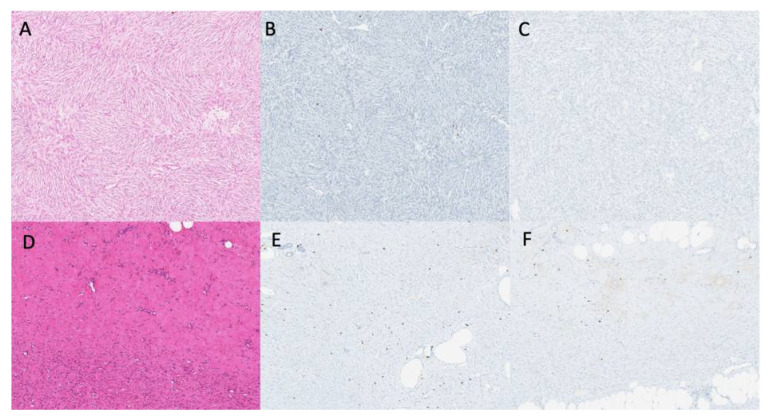
Treatment of DFSPs with IM leads to a decrease of atypical spindle cells density, induction of hyalinic fibrosis (HES coloration), and enrichment of activated T cells (CD3 and CD8 labelling) *original magnification* × 10. (**A**)—Naïve DFSP tumor with atypical spindle cells (HES coloration). (**B**,**C**)—Representative CD3 and CD8 staining on DFSP tumor before TKI treatment. (**D**)—DFSP tumor after TKI therapy showing reduction of spindle cells and fibrosis (HES coloration). (**E**,**F**)—Representative CD3 and CD8 staining on DFSP tumor after TKI treatment, showing a moderate tumoral infiltrate of activated T cells.

**Table 1 cancers-13-02224-t001:** Patients characteristics at baseline, and treatment characteristics *N* = 27 patients.

Characteristics	*N* Median (Q1; Q3) or *N* %
**Age at DFSP diagnosis**	
(years)	44.2 (33.6; 47.19)
**Sex**	
Men	15 (55.55%)
Women	12 (44.45%)
**Localization**	
Trunk	16 (60%)
Scalp + Face	9 (33%)
Limbs	2 (7%)
**Diameter:**	
Clinical (cm)	6 (5; 11.25)
Radiological (cm)	5.1 (3.5; 8.6)
**Fibrosarcomatous transformation**	
No	18 (66.67%)
Yes	9 (33, 33%)
FNCLCC grade 1	3
FNCLCC grade 2	3
FNCLCC grade 3	3
**Previous therapy for DFSP**	
Surgery without margins	17 (63%)
None	10 (37%)
**Type of neo-adjuvant TKI**	
Imatinib	22 (81%)
Pazopanib	2 (7%)
Pazopanib to Imatinib *	1 (0.04%)
Imatinib to Pazopanib **	1 (0.04%)
Imatinib/Sunitinib/Nilotinib **	1 (0.04%)
**Dose mg (median, range)**	
Imatinib	600 (400–800)
Pazopanib	700 (600–800)
**Median duration (months)**	7.02 (4.607; 12.48)
**Number of surgeries needed to obtain clear margins**	
1	10 (42%)
2	10 (42%)
3 or more	4 (16%)
Not applicable	3
**Adjuvant treatment**	
Radiotherapy	3 (11.11%) ***

* Switch because of cholestasis. ** Switch due to therapeutic failure of imatinib. *** Two patients with fibrosarcomatous transformation and one with progression with imatinib and pazopanib before surgery.

**Table 2 cancers-13-02224-t002:** Tumor evaluation.

Best Response to TKI (RECIST)	*N* (%)
**For all patients**	
Complete Response (CR)	2 (7, 69%)
Partial Response (PR)	8 (30, 77%)
Stability Disease (SD)	12 (46, 15%)
Progression Disease (PD)	4 (15, 38%)
Not available	1
**DFSP without transformation**	
Complete Response (CR)	1 (5, 55%)
Partial Response (PR)	7 (38, 89%)
Stability Disease (SD)	8 (44, 45%)
Progression Disease (PD)	2 (11, 11%)
**DFPS with transformation**	
Complete Response (CR)	1 (11, 11%)
Partial Response (PR)	1 (11, 11%)
Stability Disease (SD)	4 (44, 45%)
Progression Disease (PD)	2 (22, 22%)
Not available	1
**Last known status**	N (%)
No evidence of disease (NED)	23 (85, 19%)
PR	0 (0%)
SD with refusal of surgery	1 (3, 70%)
PD with metastasis after complete resection (R0)	1 (3, 70%)
PD before surgery	2 (7, 41%)

**Table 3 cancers-13-02224-t003:** Toxicities with TKI treatment.

Events	No. of Adverse Events				
	Total	Grade 1	Grade 2	Grade 3	Grade 4
Asthenia	14	13 (93%)	1 (7%)	0	0
Edema	9	9 (100%)	0	0	0
Nausea/vomiting/anorexia	9	6 (67%)	1 (11%)	2 (22%)	0
Diarrhea	4	3 (75%)	1 (25%)	0	0
Abdominal pain	3	3 (100%)	0	0	0
Gastroesophageal reflux	1	1 (100%)	0	0	0
Toxidermia	5	4 (80%)	1 (20%)	0	0
Aphthous stomatitis	1	1 (100%)	0	0	0
Thrombocytopenia	1	1 (100%)	0	0	0
Neutropenia	2	0	0	1 (50%)	1 (50%)
Myalgias	2	2 (100%)	0	0	0
Elevated liver enzymes	1	1 (100%)	0	0	0
Cholestasis	1	0	0	1 (100%)	0
Tachycardia	1	1 (100%)	0	0	0
Amnesia	1	1 (100%)	0	0	0
CPK increased	1	0	1 (100%)	0	0

## Data Availability

Data can be available on demand.

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
