# Peer review of "Long-Term Outcome of Neoadjuvant Tyrosine Kinase Inhibitors Followed by Complete Surgery in Locally Advanced Dermatofibrosarcoma Protuberans"

_cancers, 2021, doi:10.3390/cancers13092224_

Round 1
Reviewer 1 Report
I have the following comments/questions:
1.) please explain what you mean by supernumerary ring chromosomes.
2.) You have two different reference formats in the introduction (one in parentheses and the other as superscript).
3.) It is confusing to me as to the choice of TKI--It seems most were imatinib and then switch to pazopanib or other in the case of progression/intolerance. However, at least one patient began on pazopanib--why is this. Please explain further the rationale for which TKI was started and then used for salvage.
4.) You talk about evaluating mitotic index in the methods but do not actually mention it in the results.
5.) In the results section, you state there are 5 patients with bone invasion (page 3)--is this really invasion of the bone or abutment? Later on in the response section, there were no patients who had bone invasion on final pathology--please clarify.
6.) There were more patients that needed re-resection for positive margins than those who had clear margins after the first surgery--this says to me that the initial surgery probably was not adequate to begin with--please comment.
7.) The sentence at line 145 does not make sense to me and needs to be re-written.
8.) For Table 2, do you have the best response broken out by those with fibrosarcomatous transformation vs. pure DFSP. It would be helpful to break this out. Also, in the 2nd part of that table (last known status), it should be NED (no evidence of disease) as opposed to CR.
9.) In results, you demonstrate the histopathological response but never discuss the significance of this response.
10.) For Table 3, you should have percentages and in the key to the table state what the numbers actually represent--is it patients or events?
11.)
Author Response
Dear reviewer,
Please find enclosed a point-by-point response to your comments. We have read your detailed and constructive comments, tried to address your comments, and feel that the efforts have resulted in an improved manuscript.

Reviewer 2 Report
The manuscript addresses an interesting topic which falls within the scope of Cancers. The objective of this retrospective study was to evaluate the long-term status of patients with histologically proven locally advanced dermatofibrosarcoma protuberans who received neoadjuvant tyrosine kinase inhibitors (imatinib, pazopanib) followed by surgical treatment. The manuscript is well structured. The authors have highlighted the aims, significance and the novelty of their work and the methods used are appropriate. The results are well described showing that neoadjuvant tyrosine kinase inhibitors followed by complete surgery with micrographic analysis is an effective strategy for locally advanced dermatofibrosarcoma protuberans and the conclusions made are supported by the data presented.
However, even if this study presents the largest cohort of locally advanced DFSP without metastases and treated with neoadjuvant tyrosine kinase inhibitors with a long-term follow-up, the authors need to better highlight the novelty of their study, since other trials with similar objectives were published starting from 2005. Moreover, the results of the multicenter phase II study which included data form the same research group as the present paper were published by Kérob et al in 2010.
Furthermore, the authors must discuss the results of the present study as compared to another research of the same group, not cited in the manuscript, even if it was recently published in The Journal of Investigative Dermatology:
Delyon J, Porcher R, Battistella M, Meyer N, Adamski H, Bertucci F, Guillot B, Jouary T, Leccia MT, Dalac S, Mortier L, Ghrieb Z, Da Meda L, Vicaut E, Pedeutour F, Mourah S, Lebbe C. A Multicenter Phase II Study of Pazopanib in Patients with Unresectable Dermatofibrosarcoma Protuberans. J Invest Dermatol. 2021 Apr;141(4):761-769.e2. doi: 10.1016/j.jid.2020.06.039. Epub 2020 Sep 18. PMID: 32956651.
Author Response

(The authors gave the same response as above.)

Round 2
Reviewer 1 Report
In tables, should have decimal points instead of commas